# Enhanced Pancreatitis Detection: FAPI PET/CT Emerging Utility in Patient with Pancreatic Mass

**DOI:** 10.3390/diagnostics14151622

**Published:** 2024-07-27

**Authors:** Alain Abi Ghanem, Tamara El Annan, Nour El Ghawi, Nadine Omran, Mustafa Natout, Hazem Assi, Ali Shamseddine, Nina Saliba, Mohamad Haidar

**Affiliations:** 1Department of Diagnostic Radiology, American University of Beirut, Beirut 11-0236, Lebanon; aa277@aub.edu.lb (A.A.G.); te28@aub.edu.lb (T.E.A.); ne130@aub.edu.lb (N.E.G.); no15@aub.edu.lb (N.O.); mn113@aub.edu.lb (M.N.); ns204@aub.edu.lb (N.S.); 2Department of Internal Medicine, Hematology/Oncology, American University of Beirut, Beirut 11-0236, Lebanon; ha157@aub.edu.lb (H.A.); as04@aub.edu.lb (A.S.)

**Keywords:** FAPI PET/CT, pancreatitis, pancreatic mass, fibroblast activation protein, FDG PET/CT, diagnostic imaging

## Abstract

Fibroblast Activation Protein Inhibitor (FAPI) positron emission tomography (PET) imaging has emerged as a useful method for identifying pancreatic disorders, notably pancreatitis. Unlike Fluorine-18 fluorodeoxyglucose (FDG), FAPI uptake is directly proportional to the degree of fibrosis, making it very useful in separating pancreatic tumors from inflammation. Recent investigations have shown that FAPI positron emission tomography/computer tomography (PET/CT) can identify pancreatic inflammation with great sensitivity, providing vital diagnostic information. In this case study, a 52-year-old male with a history of Ewing sarcoma presented with epigastric pain. Pancreatitis was confirmed on a computer tomography (CT) scan showing mild fat stranding in the pancreatic body and tail, in addition to a significant increase in pancreatic head mass, necessitating further evaluation with FDG PET/CT and FAPI PET/CT, as the patient was known to have metastatic sarcoma. While FDG PET/CT revealed an avid infiltrative lesion in the duodenal/pancreatic head area, FAPI PET/CT showed diffuse uptake in the pancreatic body and tail, indicating fibroblast-mediated inflammation consistent with pancreatitis. This case demonstrates the usefulness of FAPI imaging in discriminating between pancreatic metastasis and pancreatitis, with FAPI PET/CT providing crucial diagnostic information when FDG uptake is ambiguous.

**Figure 1 diagnostics-14-01622-f001:**
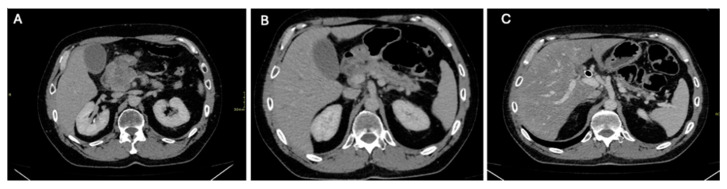
Abdominal CT Images showing pancreatic mass, pancreatic fat stranding, and post-treatment pancreatic changes: (**A**) Contrast-enhanced axial cut of the abdomen showing a large infiltrative hypoenhancing pancreatic mass measuring 6.2 × 5.7 cm with secondary mild pancreatic ductal dilatation, as well as intrahepatic and extrahepatic biliary ductal dilatation (**B**) Contrast-enhanced axial cut showing mild fat stranding at the level of the pancreatic body and tail. (**C**) Status post biliary stent. The interval significant decrease in the size of the pancreatic head mass and decrease in the encasement of the adjacent vessels. Atrophy of the pancreas. A 52-year-old male presented to the clinic for a few weeks with new-onset epigastric pain radiating to the back and the shoulder blades accompanied by constipation. He is known to have calf Ewing sarcoma, being diagnosed in 2009, and have been treated with chemotherapy and surgery. Small-cell round sarcoma recurrence was then observed in the retroperitoneal and paravertebral regions for which the patient completed chemotherapy. The patient underwent laboratory tests that revealed the following results: bilirubin total, 8 mg/dL; bilirubin direct, 6.1 mg/dL; alkaline phosphatase, 480 IU/L; alanine transaminase, 86 IU/L; Aspartate Transaminase, 60 IU/L; lipase, 232 IU/L; and amylase, 126 IU/L. To further assess the patient’s symptoms and laboratory tests, a computer tomography (CT) scan of the abdomen was conducted, revealing a pancreatic head mass causing local invasion. In addition to that, multiple hepatic metastatic lesions were observed, and mild fat stranding was noted in the pancreatic body and tail, suggestive of pancreatitis (Figure 1A,B). Taking into account the patient’s history of pancreatic metastasis, Fibroblast Activation Protein Inhibitor (FAPI) positron emission tomography/computer tomography (PET/CT) and fluorodeoxyglucose (FDG) PET/CT were requested to assess potential disease progression. The results came out as follows: On one hand, FDG PET/CT only revealed an intensely avid ill-defined infiltrative duodenal/pancreatic head lesion and multiple radiotracer avid metastatic hepatic lesions, indicating disease progression with no uptake in the body and tail of the pancreas (Figure 2A,C). On the other hand, in addition to showing increased uptake in the pancreatic mass, FAPI PET/CT revealed diffuse homogeneous intense radiotracer uptake in the pancreatic body and tail with associated fat stranding consistent with an inflammatory process, such as pancreatitis (Figure 2B,D). Thus, FDG PET/CT failed to detect this pancreatic inflammatory process identified on FAPI PET/CT. The patient received appropriate therapy including stent implantation. Follow-up CT revealed that pancreatic atrophy was noted, further reinforcing the diagnosis of pancreatitis (**C**). A few studies have highlighted the role of FAPI and its superiority to FDG in detecting tumor-induced pancreatitis [1,2]. In addition to that, many articles have demonstrated the effectiveness of FAPI compared to FDG in treating pancreatitis induced by other causes such as immunoglobulin G4 (IgG4)-related disease (IgG4-RD). For example, Shou, Xue, and colleagues [3] presented a case of IgG4-RD pancreatitis in which the pancreas and bile duct tree showed widespread inflammation on FAPI PET that was not observed on FDG PET. Luo, Pan, and colleagues [4] have also demonstrated the effectiveness of FAPI at detecting inflammatory processes in IgG4-RD. While FDG and FAPI showed high levels of radiotracer avidity in pathological parotid and submandibular glands and pulmonary lesions; however, only FAPI displayed significant uptake in the pancreas uncinate process, indicating focal pancreatitis. FDG PET is still an excellent tool for detecting inflammatory cell infiltration; however, FAPI PET may be more sensitive in tissues where fibroblast-mediated inflammation is present, such as pancreatitis [3]. These cases are in accordance with our observation, highlighting the improved ability of FAPI PET/CT to detect inflammation compared to FDG PET/CT. To the best of our knowledge, this case is the first to compare the roles of FDG and FAPI in detecting tumor-induced pancreatitis. FAPI PET/CT is, thus, effective in identifying inflammation, particularly pancreatitis, which might be missed by FDG PET. It is sensitive to fibroblast-mediated inflammation, making it a valuable complementary tool to FDG PET and enhanced CT scans [3].

**Figure 2 diagnostics-14-01622-f002:**
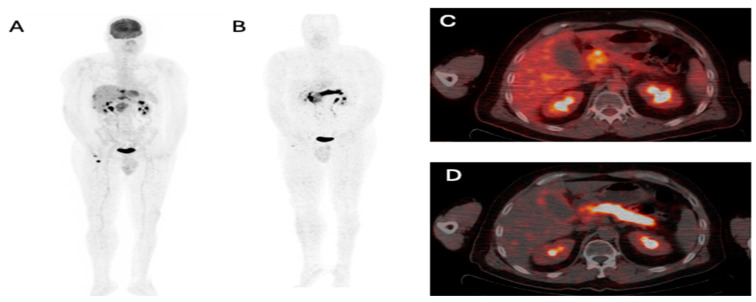
FDG PET/CT and FAPI PET/CT imaging of the pancreas: (**A**) MIP FDG PET/CT, image 32, showing focal increased radiotracer uptake at the pancreatic head and multiple scattered radiotracer avid hepatic lesions. (**B**) MIP FAPI PET/CT, image 32, showing diffuse increased radiotracer uptake in the pancreas. (**C**) Axial fused FDG PET/CT, image 33, showing focal radiotracer avid lesions at the pancreatic head, inseparable from the adjacent D2 segment of the duodenum. (**D**) Axial fused FAPI PET/CT, image 33, showing diffuse increased radiotracer uptake in the pancreas. Our findings show that using FAPI PET/CT in clinical practice can significantly improve the accuracy of diagnosing inflammatory processes. This improved ability may lead to more precise and effective treatment methods. Additionally, FAPI PET/CT’s ability to identify fibroblast-mediated inflammation may allow for earlier detection and intervention in illnesses like pancreatitis, ultimately enhancing patient outcomes and optimizing healthcare resource utilization. Future studies should aim to compare FAPI PET/CT with other diagnostic modalities. Among these, endoscopic ultrasonography-fine needle aspiration (EUS-FNA) is widely regarded as the most accurate method for evaluating pancreatic mass lesions [5,6]. Nevertheless, EUS-FNA has limitations, and in some cases, EUS-guided fine needle biopsies (FNB) may be necessary for a more accurate and thorough examination [7,8]. This study has some limitations as it is a single-case report that demonstrates the usefulness of FAPI PET/CT for diagnosing inflammation, notably tumor-induced pancreatitis. However, conducting comparative studies with other imaging methods in various clinical scenarios would offer a more comprehensive understanding of the diagnostic capabilities of FAPI PET/CT.

## Data Availability

No new data were created or analyzed in this study. Data sharing is not applicable to this article.

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
