# Peer review of "Enhanced Pancreatitis Detection: FAPI PET/CT Emerging Utility in Patient with Pancreatic Mass"

_diagnostics, 2024, doi:10.3390/diagnostics14151622_

Round 1

Reviewer 1 Report

Comments and Suggestions for Authors

1.  The PET/CT and CT pictures are clearly described. You should however include the size of the pancreatic mass in the report.

2. To improve clarity in the final publication, make sure the figures have a good resolution.

3. Provide a more thorough comparison with other research that looked into the application of FAPI PET/CT in similar circumstances.

4. Describe the possible effects on clinical practice as well as the clinical implications of these findings for the management of patients with pancreatic masses.

5. Discuss the role of Endoscopic ultrasonography and FNB in the case of pancreatic masses

6. You did not include any limitations of your study. Please analyze your research and include its shortcomings, such as its single-case design, and make recommendations for expanding on these conclusions in larger cohorts of future research. 

Author Response

Comment 1: The PET/CT and CT pictures are clearly described. You should however include the size of the pancreatic mass in the report.

Response 1: Thank you very much for your constructive feedback. The pancreatic mass was added to the manuscript (Page 2/4, line 31). 

Comment 2: To improve clarity in the final publication, make sure the figures have a good resolution.

Response 2: Figures are reviewed. Figures have good resolution. 

Comment 3: Provide a more thorough comparison with other research that looked into the application of FAPI PET/CT in similar circumstances.

Response 3: A comparison with other research that looked into the application of FAPI PET/CT in similar circumstances added (l.58-70)

Comment 4: Describe the possible effects on clinical practice as well as the clinical implications of these findings for the management of patients with pancreatic masses.

Response 4: The possible effects on clinical practice and implications have been added (l.77-81).

Comment 5: Discuss the role of Endoscopic ultrasonography and FNB in the case of pancreatic masses.

Response 5: The role of endoscopic ultrasonography and FNB has been discussed (l.82-86)

Comment 6: You did not include any limitations of your study. Please analyze your research and include its shortcomings, such as its single-case design, and make recommendations for expanding on these conclusions in larger cohorts of future research.

Response 6: Limitations have been added (l.87-90)

Reviewer 2 Report

Comments and Suggestions for Authors

Dear Authors,

Diagnostics-3073635

Enhanced pancreatitis detection: FAPI PET/CT Emerging Utility in Patient with Pancreatic Mass by Ghanem et al demonstrates Fibroblast Activation Protein Inhibitors (FAPI) positron emission tomography (PET) imaging has emerged as a useful method for identifying pancreatic disorders, notably pancreatitis. In this case study, a 52-year-old male with a history of Ewing sarcoma presented with epigastric pain. Pancreatitis was confirmed on a computer tomography (CT) scan showing a mild fat stranding in the pancreatic body and tail, in addition to a significant increase in pancreatic head mass necessitating further evaluation with FDG PET/CT and FAPI PET/CT as the patient was known to have metastatic sarcoma. While FDG PET/CT revealed an avid infiltrative lesion in the duodenal/pancreatic head area, FAPI PET/CT showed diffuse uptake in the pancreatic body and tail, indicating fibroblast-mediated inflammation consistent with pancreatitis. Additionally, this case demonstrates the usefulness of FAPI imaging in discriminating between pancreatic metastasis and pancreatitis, with FAPI PET/CT providing crucial diagnostic information when FDG uptake is ambiguous.

Comments:

1.     This is an interesting study. The authors showing one case, if the authors could show more images will increase the quality of the manuscript.

2.     iThenticate report is 25%. It will be nice to see below 15%.

Comments on the Quality of English Language

Minor edits required

Author Response

Comment 1: This is an interesting study. The authors showing one case, if the authors could show more images will increase the quality of the manuscript.

Response 1: Thank you for your input. We think we have included all pertinent images. Please let us know if any specific image is needed. 

Comment 2:  iThenticate report is 25%. It will be nice to see below 15%.

Response 2: Thank you for your comment. We have reviewed our manuscript and ran our manuscript on a website that detected no plagiarism. Can you please point out the sections that need modifications? 

Reviewer 3 Report

Comments and Suggestions for Authors

This manuscript shows that PET/CT can be used to image the pancreatic tumor non-invasively. The abdominal CT scan as well as fusion of PET/CT imaging have shown the capability to image the pancreatic tumor and the radiotracer.

Comments on the Quality of English Language

-

Author Response

Comment 1: This manuscript shows that PET/CT can be used to image the pancreatic tumor non-invasively. The abdominal CT scan as well as fusion of PET/CT imaging have shown the capability to image the pancreatic tumor and the radiotracer.

Response 1: Dear Reviewer, thank you very much for your feedback. Manuscript was reviewed to improve its quality. Kindly let us know if any further modifications are needed.